# Embracing Contradiction: Theoretical Inconsistency Will Not Impede the Road of Building Responsible AI Systems

**Gordon Dai***
New York University
td2568@nyu.edu

**Yunze Xiao** *
Carnegie Mellon University
yunzex@andrew.cmu.edu

## Abstract

This position paper argues that the theoretical inconsistency often observed among Responsible AI (RAI) metrics, such as differing fairness definitions or trade-offs between accuracy and privacy, should be embraced as a valuable feature rather than a flaw to be eliminated. We contend that navigating these inconsistencies, by treating metrics as divergent objectives, yields three key benefits: (1) Normative Pluralism: maintaining a full suite of potentially contradictory metrics ensures that the diverse moral stances and stakeholder values inherent in RAI are adequately represented; (2) Epistemological Completeness: using multiple, sometimes conflicting, metrics captures multifaceted ethical concepts more fully and preserves greater informational fidelity than any single, simplified definition; (3) Implicit Regularization: jointly optimizing for theoretically conflicting objectives discourages overfitting to any one metric, steering models toward solutions with better generalization and robustness under real-world complexities. In contrast, enforcing theoretical consistency by simplifying or pruning metrics risks narrowing value diversity, losing conceptual depth, and degrading model performance. We therefore advocate a shift in RAI theory and practice: from getting trapped by metric inconsistencies to establishing practice-focused theories, documenting the normative provenance and inconsistency levels of inconsistent metrics, and elucidating the mechanisms that permit robust, approximated consistency in practice.

## 1 Introduction

The current practice of Responsible AI is built on metrics. Fairness audits invoke demographic parity [1], equalized odds [2], or counterfactual consistency [3]; privacy claims are based on $\varepsilon$-differential privacy [4]; robustness tests span distribution shift [5], adversarial risk [6], and calibration error [7]. Each metric quantifies an abstract virtue into numbers to enable evaluation and optimization. Yet beneath this quantification enterprise lies a fundamental puzzle: **many of these metrics are mathematically incompatible**. Classical impossibility theorems show, for example, that no nontrivial predictor can simultaneously satisfy three most common fairness definitions except in degenerate cases of perfect prediction or equal base rates [8, 9, 10]. Similar trade-offs plague accuracy versus privacy [4, 11], political neutrality versus informativeness [12], and interpretability versus expressive power [13].

Conventional wisdom treats these contradictions as *bugs to be patched*: choose a single "right" metric [14, 2] or derive consistency constraints [15, 16]. We take the opposite stance. Drawing on work in moral philosophy, Human-Computer Interaction (HCI), and multi-objective optimization, we argue that **theoretical inconsistency is a feature, not a flaw**. The value of preserving theoretically

---

*Gordon Dai and Yunze Xiao contribute equally to this work

39th Conference on Neural Information Processing Systems (NeurIPS 2025) Position Paper Track.

inconsistent metrics emerges across three dimensions: Normatively, they encode distinct commitments from diverse social groups, essential for pluralistic alignment. Epistemologically, they enable a richer understanding of complex Responsible AI concepts. Practically, jointly optimizing these conflicting objectives acts as an implicit regularizer, steering learners away from brittle, single-metric shortcuts and towards solutions that generalize under realistic uncertainty [17, 18].

Overall, this paper makes following three contributions:

1. We formalize two kinds of metric inconsistency—*intra-concept inconsistency* (variants of the same ideal collide) and *inter-concept trade-off* (distinct ideals compete)—and illustrate how each kind can project a complicated picture when applied to reality, with several canonical examples.

2. We synthesize evidence that inconsistent objectives improves ethical coverage, conceptual understanding, and out-of-sample performance, linking insights from optimization theory, Pareto-front geometry, Rashomon set exploration, and the Goodhart's Law.

3. We introduce a practical protocol that replaces fixed tolerances with transparent *documentation of inconsistency*: pre-registering the inconsistency functional and summaries, publishing Metric Provenance Sheets for each metric, and reporting empirical inconsistency along four axes (magnitude, directionality, locality, sensitivity) with uncertainty, stakeholder review, and drift monitoring via versioned SPHERE cards.

In short, we invite the community to embrace contradiction as the necessary price and the promise of building Responsible AI systems that serve a pluralistic world. In the remainder of this paper, we identify two types of inconsistencies in Section 2 to provide a conceptual foundation for our central claim: theoretical contradictions between metrics—far from being flaws—serve practical and normative purposes, followed by Section 3 where we present normative and empirical support for this position from the aforementioned dimensions. Next, Section 4 outlines actionable recommendations for theorists, tool builders, and regulators for engaging with theoretical contradiction. Finally, in Section 5 we synthesize our responses to alternative views on our position, before we conclude the argument in Section 6.

## 2 Conceptual Framework: Two Forms of Metric Inconsistency

To make sense of the tension between Responsible AI metrics, we define two formal types of inconsistency that underlie these conflicts. The first, which we term *intra-concept inconsistency*, occurs when multiple metrics derived from the same normative concept (e.g., fairness) conflict with each other (see Definition 1). The second, *inter-concept inconsistency*, arises when optimizing for one desirable metric (e.g., accuracy) degrades performance on another (e.g., privacy or fairness) due to structural trade-offs (see Definition 2). Below, for each of these two inconsistencies, we will illustrate with canonical examples and show how, in each case, the inconsistency in theory formed a conversation with empirical results that more or less suggests an approximate consistency.

---

**Definition 1: Intra-concept inconsistency**

Let $\mathcal{H}$ be a hypothesis space (all possible models) and $\mathcal{A} = \{a_1, \ldots, a_n\}$ where $a_i : \mathcal{H} \to \{0, 1\}$ are Boolean metrics that all purport to measure the normative concept *same A* (e.g. fairness). 0 denotes unsatisfied and 1 denotes satisfied. We say $\mathcal{A}$ is inconsistent if

$$\nexists h \in \mathcal{H} \text{ such that } \forall i : \ a_i(h) = 1,$$

unless a trivial edge case holds (e.g: perfect prediction, identical base rates).

- - - - - - - - - - - - - - - - - - - - - - - - - - - - - - - - - - - - - - - - - - - - - - - - - - - - - - - - - - -

**Interpretation.** No single model can make *all* fairness metrics "satisfied" at once except in degenerate situations.

---

## 2.1 Fairness

**Fairness illustrates a classic case of inconsistency between concepts, where multiple metrics derived from the single normative concept of algorithmic fairness cannot be simultaneously satisfied.** Kleinberg et al. [8] demonstrated that three commonly used fairness metrics, equalizing calibration within groups, maintaining balance for the negative class, and maintaining balance for the positive class, could not be concurrently satisfied across multiple groups, with only two exceptions [8]. These exceptions occurred: (1) when the algorithm achieved perfect prediction or (2) when there was no prevalence difference between the groups. Chouldechova [9] formulated a similar impossibility result, expressing it as a relationship between *Predictive Positive Value* (PPV), *False Positive Rate* (FPR), *False Negative Rate* (FNR), and *prevalence* ($p$), as shown in Equation 1:

$$\text{FPR} = \frac{p}{1-p} \cdot \frac{1 - \text{PPV}}{\text{PPV}} \cdot (1 - \text{FNR}) \tag{1}$$

More recently, Bell et al. [19] engaged with this theoretical impossibility: Instead of considering the perfectly fair case among multiple metrics, by slightly loosening the constraint from *zero* disparity to *minimum* disparity, one would find plenty of models that were approximately fair with respect to these theoretically inconsistent metrics [19]. Their empirical studies on 18 real-world datasets revealed that theoretical impossibility results often overstate practical trade-offs. This perspective gained support from Wick et al. [20], who demonstrated that carefully engineered feature representations could mitigate fairness trade-offs, and Liu et al. [21], who argued that impossibility results often stemmed from implicit assumptions about data generation processes.

## 2.2 Political Neutrality

**Like fairness, political neutrality exemplifies intra-concept inconsistency, where multiple interpretations derived from this single normative concept cannot be simultaneously satisfied.** Drawing from political philosophy, John Rawls argued that *procedural*, *aim* and *effect neutrality* could not be jointly satisfied [22, 23]. In particular, the third sense was "undoubtedly impossible" and "futile to try to counteract": educational institutions inevitably tilted the social climate, so we must abandon the neutrality of effect [23]. Joseph Raz made a similar point on a parallel concept: comprehensive neutrality, being neutral with respect to the ideals people will adopt in the future, was also unattainable in practice, given that any serious political morality unavoidably shaped the comparative fortunes of conceptions of the good [24].

Yet Raz also argued that neutrality "can be a matter of degree" [25]. Recent empirical work by Fisher et al. [12] inherited and operationalized Raz's "degrees of neutrality" idea by proposing eight mathematically formalized techniques for approximating political neutrality across three levels: output level (refusal, avoidance, reasonable pluralism, output transparency), system level (uniform neutrality, reflective neutrality, system transparency) and ecosystem level (neutrality through diversity). Their evaluation of 9 LLMs in 7,314 political queries demonstrated that these approximations could be practically implemented with measurable trade-offs between utility, safety, fairness, and user agency. For example, while refusal techniques achieved 100% safety scores, they scored poorly on utility, while reasonable pluralism maintained high fairness but risked information overload.

---

**Definition 2: Inter-concept trade-off**

Let $\mathcal{H}$ be a hypothesis space and $A, B : \mathcal{H} \to \mathbb{R}_{\geq 0}$ be *different* metrics (e.g. accuracy and demographic parity or loss of privacy). There is an $(A, B)$ trade-off if

$$\sup_{h \in \mathcal{H}:B(h) \leq b} A(h) \quad < \quad \sup_{h \in \mathcal{H}} A(h) \quad \text{for some } b < \sup_h B(h).$$

- - - - - - - - - - - - - - - - - - - - - - - - - - - - - - - - - - - - - - - - - - -

**Interpretation.** Constraining $B$ (say, requiring loss of fairness $\leq b$ or privacy $\varepsilon \leq b$) reduces the maximum achievable $A$ (say accuracy) below its unconstrained optimum.

---

## 2.3 Accuracy–Fairness

An illustrative inter-concept trade-off involves accuracy and fairness. From an information theory perspective, Zhao & Gordon [11] derived lower bounds to show that satisfying *independence-based* parity notions such as demographic (statistical) parity would impose extra error when group base rates differed. Specifically, for binary class labels $Y \in \{0, 1\}$ and a protected attribute $A \in \{0, 1\}$, given $\text{Err}_g(h)$ denotes the misclassification rate of hypothesis $h$ on group $A = g$, they proved that

$$\text{Err}_0(h) + \text{Err}_1(h) \geq \big|\Pr(Y = 1 \mid A{=}0) - \Pr(Y = 1 \mid A{=}1)\big|. \tag{2}$$

Thus, when the base-rate gap on the right-hand side is large, at least one group must incur a proportionally large error.

The empirical findings paint a more nuanced picture. Hardt et al. [26] demonstrated that, via post-processing optimization, *error-rate-matching* criteria—such as equalized odds or equality of opportunity—can be satisfied with negligible loss in overall accuracy. Rodolfa et al. [27] corroborated this across several public-policy tasks and observed virtually no reduction in precision after post-hoc mitigation of recall disparity. Furthermore, Li et al. [28] showed that enforcing their causal-path fairness constraint can even *improve* accuracy.

The key insight is that the **accuracy–fairness trade-off is not universal but depends on the particular fairness metric**. Independence-based metrics tend to incur an accuracy cost, whereas error-rate-matching criteria, calibration, or certain causal formulations can often be achieved at little or no cost. Analogous debates have arisen over trade-offs between accuracy–interpretability [13, 29] and accuracy–privacy [30].

## 3 The Value of Inconsistent Metrics

Reviewing three of the aforementioned case studies, one might naturally ask: given the theoretical inconsistencies of Responsible AI metrics, does this suggest that the underlying concepts such as "fairness" and "neutrality" behind these metrics are *ill-defined*? Do these concepts such as "fairness" and "neutrality" remain meaningful and valuable in Responsible AI?

We note that this question can be generalized to: if a goal is contradictory (here making the machine learning model political neutral), should one pursue it (to optimize the model for it)? In other words, should one only pursue different goals that are consistent with each other? **Our position is that we should embrace this inconsistency**.

In this section, we emphasize the value of preserving theoretically inconsistent metrics by ensuring that these inconsistencies serve three key purposes: 1) Normatively, they uphold value pluralism: each metric captures a distinct moral stance, ensuring that diverse stakeholder perspectives remain visible. 2) Epistemologically, these inconsistent metrics better preserves the information in the underlying concept. 3) Practically, conflicting metrics act as regularizers, guiding models toward more robust and generalizable behavior under real-world complexity. Rather than impeding progress, inconsistency enables both ethical inclusivity and technical resilience.

### 3.1 Inconsistent metrics encode the (unfortunately) inconsistent human values required in pluralistic alignment.

In AI Alignment, pluralistic alignment is an approach that acknowledges and embraces the diversity of human values, perspectives, and preferences rather than attempting to align AI systems with a single, universal set of values [31, 32, 33]. The core premise of pluralistic alignment is that there isn't one "correct" set of human values that AI systems should adopt. Instead, it recognizes that different cultures, communities, and individuals have varying and sometimes conflicting values, and that AI systems should be designed to accommodate this diversity, rather than propagating bias and systematic injustice [34, 35, 36, 37, 38].

From the famous proverb, "There are a thousand Hamlets in a thousand people's eyes", it is natural for a concept to be understood differently among people [39], social groups [40], and culture [41]. There is no exception to core concepts in Responsible AI such as "fairness", "privacy", and "political neutrality". Each metric of the concept exactly represents one way of understanding the concept

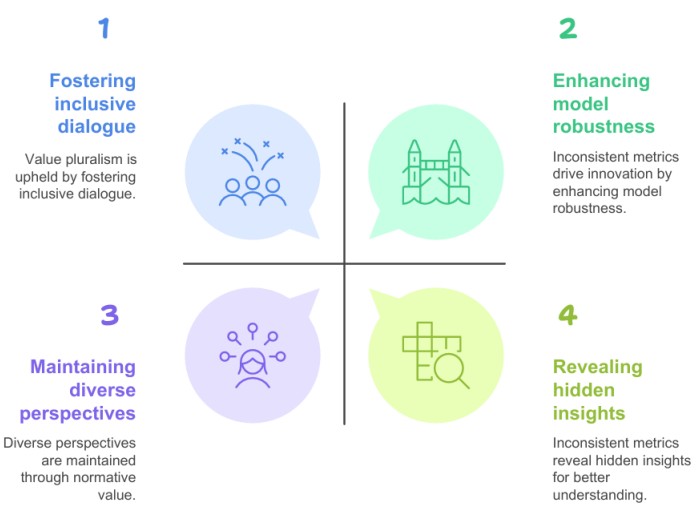

**Value of Inconsistent Metrics**

**1**
**Fostering inclusive dialogue**

Value pluralism is upheld by fostering inclusive dialogue.

**2**
**Enhancing model robustness**

Inconsistent metrics drive innovation by enhancing model robustness.

**3**
**Maintaining diverse perspectives**

Diverse perspectives are maintained through normative value.

**4**
**Revealing hidden insights**

Inconsistent metrics reveal hidden insights for better understanding.

Figure 1: An Overview of The Values in Theoretical Inconsistency

by an individual or a social group. In turn, people from different social groups may have divergent conception of a single concept. For example, psychology literature has shown that people from individualist cultures tend to favor the rule of equity and the distributive principle of equity, while people from collectivist cultures emphasize equality and need-based rules, especially with members of the group [42, 43, 44]. This might suggest that if a model is intended to be applied to a society where people from divergent social backgrounds mix, to ensure pluralistic alignment, multiple fairness metrics including demographic parity and individual fairness or equal opportunity are needed.

On the other hand, diversified human values are always inconsistent with each other. Isaiah Berlin argued that fundamental human values are inherently pluralistic and sometimes cannot be reconciled theoretically [45]. As Berlin argued, **the incompatibility of values does not make them less valid** - it is a natural condition of human life that we must navigate. Value pluralism offers an alternative to both moral relativism (all values are equally valid) and moral absolutism (only one set of values is true)[12]. In this view, the lack of theoretical consistency between Responsible AI metrics isn't a flaw but reflects the genuine plurality of human values. Thus, if one finds several metrics of a single Responsible AI concept inconsistent (i.e: facing intra-concept inconsistency as defined in 1), this suggests that the diversified understandings of that concept are inconsistent.

Therefore, if one manages to fix the inconsistency by simply considering a subset of the set of all metrics around the concept, it would harm the goal of pluralistic alignment, as it made some of the respective understanding of the concept behind the deleted metric underrepresented in the evaluation and optimization. For instance, in the case of fairness, if one focuses on solely optimizing for Predictive Positive Value and False Negative Rate, the idea of fairness as understood by people who proposed False Positive Rate is underrepresented. Therefore, optimizing machine learning models with respect to these inconsistent metrics preserves the plural human values, thus helping the agenda of pluralistic alignment.

### 3.2 Inconsistent metrics better preserves information of the underlying concept.

Besides the ethical dimension of inconsistent metrics, we further argue that epistemologically, preserving inconsistent metrics around a concept is to best preserve the information contained within that concept.

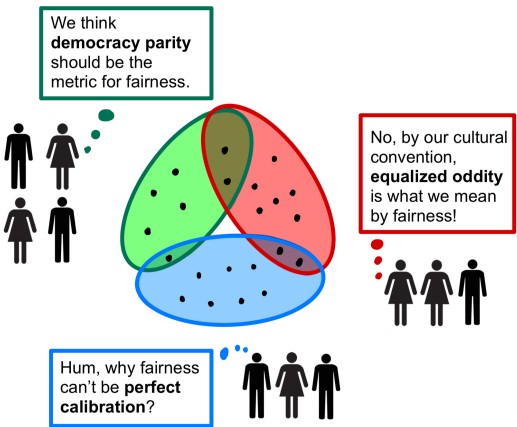

Figure 2: Represented as clusters in different colors, different groups have different ideas about what "fairness" means and formalized them into divergent metrics. By the Impossibility Theorem of Fairness, no models, as represented by black dots, can satisfy all of the three formalized metrics of fairness.

Wittgenstein's analysis of "family resemblance" terms shows that many everyday and moral concepts such as "game", "fair", "neutral" do not have one set of necessary and sufficient conditions. There is no one property (or fixed list of properties) that every "game" has and that only games have. Some games are competitive, some are cooperative; some require skill, others chance; some have rules, others only conventions. Instead, different games share different combinations of features. Chess and soccer both have rules, but chess and tag both involve winning–losing. Each pair overlaps on some trait, but there's no single trait common to all [46].

A formal metric of a concept selects one cluster of it and thereby projects only part of the resemblance network. Any such projection might be useful, but is inevitably a "rational reconstruction" that sacrifices some of the original content [47]. For instance, to capture human "intelligence", psychologists proposed several metrics, such as the genuine Intelligence Quotient (IQ) [48], Emotional Quotient (EQ) that captures social and emotional understanding [49, 50], Gardner's Multiple Intelligences that capture linguistic, spatial, musical, and other domains [51], and Practical Intelligence that captures real-world problem solving [52, 53]. Each of these tests captures a limited cluster of what we mean by "intelligent." Similarly, for concepts in responsible AI such as political neutrality, each of the breakdown dimension such as "refusal", "avoidance", or "reflective neutrality" as specified in [12] captures a limited cluster of "political neutrality".

Hence, **the best attempt to capture a concept is to incorporate all the possible metrics (which are potentially inconsistent) into the evaluation**, so that all the possible clusters are getting involved, and therefore the original content is best preserved. One would have a better understanding of their "intelligence" by considering their performance on all of the possible tests; one would have their best capture of the degree of political neutrality of an AI model if its performance on all the possible dimensions of neutrality is considered. Conversely, if one manages to fix the inconsistency by simply considering a subset of the set of all metrics around the concept, it will inevitably lead to the loss of some information contained in the concept.

## 3.3 Metric Inconsistency Has Potential Practical Values

Furthermore, we emphasize the potential practical value of metric inconsistency. The conventional wisdom that *one metric suffices* is often fragile in optimization practice: once a single score becomes the sole optimization target, models exploit idiosyncrasies of the training distribution—a phenomenon formalized by Goodhart's Law [54]. Here, we show that maintaining and jointly optimizing several conflicting objectives counteracts such specification overfitting. **It acts as an *implicit regularizer* that enhances out-of-sample accuracy and robustness**. Three complementary mechanisms are identified and supported by theory and evidence below.

**Mechanism I: Gradient Conflict as Implicit Regularization** In multi-objective learning, when optimizing $\mathcal{L}(x; \theta) = \sum_i \lambda_i L_i(x; \theta)$, objectives are said to conflict when the angle between their gradients exceeds 90°[18]. This misalignment bounds the effective gradient norm via the parallelogram law and prevents any one loss from dominating updates[55]. Crucially, these conflicting gradients introduce *implicit regularization*: unlike stochastic methods such as dropout, this regularization arises from meaningful tension between goals and narrows the train–validation gap [18, 56]. Algorithms like PCGrad [57] and CAGrad [58] leverage this structure to promote task-agnostic feature learning. For example, in NYUv2 semantic-depth co-training, PCGrad improves mean IoU by 2.3 percentage points while also reducing depth RMSE by 0.04m [57]. Theoretically, when the $L_i$ are $L$-Lipschitz and of VC-dimension $d$, the generalization bound under joint optimization degrades only by $\mathcal{O}(\sqrt{k})$—a modest price for increased robustness [59].

**Mechanism II: The Existence of Pareto Fronts and the Rashomon Set** Since conflicting objectives rarely admit a unique minimizer, they may induce a *Pareto front* with a $\varepsilon$-optimal region $\mathcal{R}_\varepsilon = \{\theta \mid \exists \theta' : L_i(\theta) \leq L_i(\theta') + \varepsilon, \forall i\}$. This coincides with the notion of *Rashomon set*, defined as a set of near-optimal models with some $0 << \epsilon << 1$ accuracy loss [60, 61, 62]. Here, a larger $\mathcal{R}_\varepsilon$ yields two tangible benefits that can now be computed. First, it allows practitioners to *swap* models to satisfy downstream constraints such as fairness, interpretability, or energy budgets, without retraining or sacrificing accuracy. This flexibility is enabled by recent algorithmic advances, such as TreeFARMS and the GAM Rashomon Set algorithm, which make it feasible to enumerate near-optimal models across the Pareto front in minutes [63, 64]. Second, the functional diversity within $\mathcal{R}\varepsilon$ enhances the ensemble and majority vote strategies, improving the robustness to adversarial perturbations and distributional shifts [65]. These practical benefits are supported by both classical methods like NSGA-II [66] and modern enumeration techniques for decision trees [29], which show that Pareto-style search can reliably uncover diverse, near-optimal hypotheses—transforming ambiguity in model selection into a powerful tool for structured flexibility.

**Mechanism III: Complementary Metrics Block Shortcut Features** Requiring simultaneous performance on inconsistent metrics forces the model to abandon brittle shortcut features. For instance, in medical imaging, injecting a differential-privacy (DP) loss caps memorization, thereby *improving* external-hospital AUC by 5% despite a 2% decline on the internal test set [30]. In text classification, optimizing both sentiment accuracy and gender independence removes name-related artifacts, raising cross-domain $F_1$ [26]. Empirically, errors on one metric often flag spurious correlations exploited by the other, creating a form of *cross-metric debugging* unavailable in single-objective training.

From the above three mechanisms, inconsistent metrics are not an impediment but a *safeguard*: Especially in high-stakes domains—health, finance, criminal justice—joint optimization delivers a principled trade between slightly lower headline scores and substantially higher reliability.

## 4 Recommendations and Future Directions

**Advancing Practice-Driven Theories on Responsible AI.** A question yet to discussed in Section 2 is the gap between the inconsistencies in theory and the more complicated empirical results: Given that in practice, one can solve the inconsistency by loosing constraints, do the theoretical results really matter? How should we build theories on Responsible AI topics?

*The gap between advanced theoretical results and everyday practices in physics*

- - - - - - - - - - - - - - - - - - - - - - - - - - - - - - - - - - - - - - - - - - - - - - - - - - - -

In the physical sciences, a conceptual rift exists between the everyday phenomena we observe and the theoretical frameworks that best describe them. Classical physics provides a reliable account of the world we interact with — falling objects, mechanical systems, fluid dynamics, even though it is formally superseded by modern theories such as relativity and quantum mechanics. The latter offer more precise and comprehensive explanations of the natural world at microscopic scales and extreme velocities [67], yet their insights rarely surface in the texture of daily experience. As Thomas Kuhn noted in *The Structure of Scientific Revolutions*, older paradigms often persist as practical tools even after they are theoretically outdated [68]. Similarly, although Einstein's relativity and quantum theory revolutionized physics, classical mechanics remains foundational

> in engineering and daily applications due to its predictive adequacy within its domain [69, 70]. Classical physics endures not because it is strictly true, but because it *fits* better to everyday problems.

We highlight that the gap between theoretical inconsistencies and practical consistencies of Responsible AI metrics does not suggest that *any* theories on Responsible AI are not needed. Instead, it pushes researchers to **build theories that fit everyday Responsible AI practices**. While current theoretical formulations such as the Impossibility Theorem of Fairness often focus on optimality, the models that work well in practice are frequently sub-optimal, approximate, and constrained. What Responsible AI currently lacks is an equivalent to "classical physics" – a theoretical framework that adequately explains everyday cases of Responsible AI Practices. We should develop responsible AI theories that fit the problem context, rather than demanding universal consistency.

One of the very recent theoretical developments alongside this agenda is the theories on the Rashomon set, which theorizes properties of models with near-optimal performance, a practically feasible setting. In a recent work by Dai et al. [71], researchers explored several theoretical properties of the Rashomon set. In particular, they derived that the asymptotic size of the Rashomon Set (and thus the possibility of finding the desired models) grows exponentially with $\sqrt{\epsilon}$. This entails that in practice, a company searching for fairer models within the Rashomon set should use the largest error tolerance acceptable to their business. We argue that in the field of Responsible AI, practice-driven theories such as this should be the direction of future theoretical works.

**Documenting Normative Assumptions and Empirical Levels of Inconsistency.** We recommend all Responsible AI evaluations to document (i) the *normative assumptions* behind each metric and (ii) the *observed levels of inconsistency* among metrics. We propose (i) builds on the success of Model Cards [72], which aimed to foster transparency in model reporting by detailing intended use cases, evaluation conditions, and ethical considerations and Data Statements [73], which provided schema for documenting data set creation rationale, demographic coverage, and limitations to help practitioners understand what system behavior can (and cannot) be trusted to generalize. One can adapt these two prototypes into a concise "Metric Provenance Sheet" that states what aspect of the concept of interest each metric means to measure, what it omits, and which stakeholder values it manages to reflect [41, 74]. For (ii), one can report the empirical *level of inconsistency* without endorsing the tolerance. This level of inconsistency includes magnitude (e.g., gaps, ratios, dominance gaps on the Pareto frontier), directionality (whether metrics are mostly aligned or mostly opposed and under which modeling/data regimes), locality (which subpopulations or slices drive the divergence), and sensitivity (how levels change under alternative labeling, features, or constraints) [75]. This systematic documentation makes theoretically inconsistent metrics *auditable and comparable* across deployments, clarifies trade-offs without pre-judging them, and supports transparent negotiation of priorities among stakeholders.

**Testing Human–Metric Interaction Empirically.** To validate the practical relevance of theoretical inconsistency, we call for empirical studies involving stakeholders such as users, domain experts, and regulators in the negotiation and selection of metrics[41]. Previous work in HCI and Responsible AI has shown that participatory methods effectively capture diverse notions of fairness and stakeholder-specific priorities. For example, Cheng et al. [76] involved child-welfare practitioners in defining fairness criteria, revealing substantial variation in ethical interpretations. Zhu et al. [77] and Nakao et al. [78] developed interactive interfaces that enabled participants to navigate algorithmic trade-offs and identify the types of guidance needed for informed decision-making. Future studies should examine how people interact with pluralistic evaluation tools, make trade-offs between conflicting objectives, and interpret explanations of inconsistency. Such insights will inform the design of more intuitive interfaces [79], inclusive optimization strategies, and accountable AI policies.

## 5  Response to Alternative Views

**This plan sounds great, but practically speaking, conflicting metrics will confuse end-users and regulators!**

*Response:* Pluralistic evaluation can appear confusing, so we anchor trust with process guarantees. We pre-register the metric set, aggregation rules, and tolerance bands before model selection; pair

each metric with an Evaluation Card that states intent, assumptions, limits, and failure modes; and publish a profile dashboard with bands plus a change log similar to MMLU-MMLU-ProX [80]. We implement these cards using SPHERE in the appendix, which organizes reporting across subject, process, actor, time, and robustness [74]. Metrics serve as guardrails that support comparability, auditability, and enforceability, complemented by qualitative analysis and stakeholder input.

For regulation, we replace single-number compliance with profile-based compliance. Regulators keep minimum protections as floors on a small mandatory subset, then bound residual tensions with pre-registered tolerance bands over the full profile. This preserves clear pass–fail decisions while making trade-offs explicit and bounded. The approach aligns with risk-managed practices such as EU AI Act technical documentation and post-market monitoring and the NIST AI RMF Map–Measure–Manage functions. Empirical results show that approximate satisfaction of jointly inconsistent metrics within small tolerances is often attainable [19, 71, 81].

Operationally, we use a metric governance plan: pilot studies to propose bands with uncertainty, stakeholder review to finalize them, conflict-aware training with per-objective ablations, explicit cost reporting, and hard overrides where safety floors cannot be traded for utility. Results are communicated through public dashboards and auditable logs, and independent audit panels review systems holistically so no single metric dominates. Although our examples focus on fairness and privacy, the same floors-and-bands protocol applies to explainability and adversarial robustness[82], with SPHERE-based templates provided in the Appendix A [74].

**Be careful here! It is the diversity of metrics, not the inconsistency of metrics, that helps pluralistic alignment.**

*Response:* We acknowledge that it is not a logical necessity that inconsistency *entails* plurality. And with no doubt, we wish all the metrics to be consistent with each other while not harming plurality: all concepts do not suffer from any internal consistencies so that we can possibly achieve zero loss for all the metrics, and there will be no trade-offs among any pair of concepts central in Responsible AI. However, it is believed that inconsistent perspectives and incommensurable values are ubiquitous among real social interactions, as we established earlier in Section 3.1. In practice, a multitude of metrics will almost inevitably exhibit inconsistencies. Therefore, attempts to enforce consistency by reducing the number of metrics inherently sacrifice valuable diversity and plural representation. Hence, the viable approach to preserve plurality is to preserve inconsistent metrics.

**I see your point on the value of inconsistent metrics, but instead of considering all these metrics, shouldn't one pick a metric that works best for the application tasks, which does not involve inconsistency?**

*Response:* We indeed admit that one should choose the metric that works best for their related application tasks. However, as we illustrated in Section 3.1 there are always cases in which the targeted population shares conflicting perspectives, which requires the "best fit" evaluation method to be itself incorporating plural perspectives. This means that multiple theoretically inconsistent metrics are still needed.

Besides, even for tasks that do not assume to represent a plural perspective, inter-concept trade-off as defined in 2 remains present. For example, Differential Privacy is legally mandated [83] and ethically required to protect patient identity, yet it is exactly the privacy metric that has a trade-off with task precision (ROC-AUC) [30].

Furthermore, the effectiveness of each metric in evaluating the respective domain is not static. Some metric might be good for the application at the beginning, but, by Goodhart's law, when a good metric becomes a target to explicitly optimize for, it ceases to be a good one. Yet, if multiple (inconsistent) metrics are present in evaluation and optimization, as we established earlier in Section 3.3, the model will have less chance of suffering from the negative impacts of Goodhart's law.

**To what extent can ethics in machine learning be assessed with metrics, and how meaningful is such assessment?**

*Response:* Metrics are useful but limited. We use them as guardrails that enable comparability, auditability, and enforcement, not as final truths. Ethical assessment should report a plural profile of metrics with pre-registered floors and tolerance bands, documented via provenance or SPHERE-style evaluation cards that state intentions, assumptions, and limits. Numbers must be paired with qualitative methods and stakeholder input to capture context, harms, and value trade-offs. In short,

metric-based evaluation is possible and meaningful for accountability, provided it is plural, transparent, and embedded in participatory review.

On the other hand, multi-objective optimization introduces computational overhead and potential training instability (e.g., gradient conflict), and pluralistic auditing increases organizational burden. These costs can be bounded by employing conflict-aware update rules and early-stopping tied to floors, using Pareto/Rashomon search to identify near-optimal models with minimal utility loss, and standardizing evaluation through reusable artifacts. We will discuss this in Appendix B. While nontrivial, these mitigations render pluralistic evaluation operational at acceptable cost and with clear accountability.

## 6    Conclusion

Rather than viewing metric inconsistencies as flaws to fix, we argue they are essential features of responsible AI systems. Inconsistent metrics preserve diverse stakeholder values, capture the full complexity of ethical concepts, and act as implicit regularizers that improve model generalization. Empirical evidence shows that theoretical impossibilities become manageable trade-offs when perfect satisfaction is relaxed to approximate satisfaction within near-optimal solution spaces. The field should therefore shift from seeking universal consistency to a transparent protocol for *documenting and navigating* inconsistency. Embracing contradiction means embracing the complexity of human values; progress in responsible AI requires systems that can hold multiple truths while making trade-offs explicit, traceable, and auditable.

## Acknowledgments

We would like to thank Dr. Daniel Neill, Dr. Emily Black, and Dr. Michael Strevens for their wholehearted support in helping us brainstorm the paper at the initial stage. The authors also thank Dr. Wenyue Hua, Xuanyi Wang, Tianjian Liu, Qi Wei, and Wilson Wei Xin for reviewing our draft and providing helpful feedback.

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

# A  Document empirical inconsistency (using SPHERE as an example)

## A.1  Formalizing Reported Metric

This section formalizes the way to jointly analyze multiple metrics discussed in the second paragraph of Section 4. We introduce four possible measures: magnitude, directionality, locality, and sensitivity, which together capture how metrics diverge, align, concentrate, and vary under changes.

Let a model $h$ be evaluated with $K$ metrics. Let $\mathbf{M}(h) = (M_1, \ldots, M_K) \in \mathbb{R}^K$ denote the system-level metric profile, where each $M_j$ is the model's score on each metric (e.g: accuracy, demographic parity).

**Magnitude** quantifies how far the metrics are from agreeing at the system level. Possible measures of the inconsistency width function $\psi : \mathbb{R}^K \to \mathbb{R}_{\geq 0}$ are

$$\text{Gap} \;=\; \max_{1 \leq j \leq K} M_j \;-\; \min_{1 \leq j \leq K} M_j \tag{3}$$

$$\text{Ratio} \;=\; \frac{\max_j M_j}{\min_j M_j} \quad \text{(when all } M_j > 0\text{)} \tag{4}$$

**Directionality** captures whether metrics tend to move together or part when, for example, data or model architecture changes. Consider a set of regimes $\mathcal{R}$ (for example model variants, training

objectives, or data sets). For each regime $r \in \mathcal{R}$, compute the system-level profile $\mathbf{M}^{(r)}$ and define pairwise concordance over regimes:

$$\rho_{ij} = \text{corr}\big(\{(M_i^{(r)}, M_j^{(r)}) : r \in \mathcal{R}\}\big), \tag{5}$$

$$\text{Align} = \frac{2}{K(K-1)} \sum_{1 \leq i < j \leq K} \text{sign}(\rho_{ij}). \tag{6}$$

Hence, values near $+1$ indicate mostly aligned metrics and near $-1$ indicate mostly opposed metrics.

**Locality** identifies *where* inconsistency concentrates with respect to a contextual attribute. Let $P$ be a discrete attribute on which the target metrics are defined (e.g., groups of $P$ when assessing fairness), and let $Z$ be any contextual attribute (discrete or continuous). Fix $K$ nonnegative metrics $\{M_1, \ldots, M_K\}$ computed with respect to $P$ *within* any subset $S$ of the domain of $Z$ (for discrete $Z$, $S$ is a set of categories; for continuous $Z$, an interval or window). Write the slice profile as $\mathbf{M}^{(S)}(h) \in \mathbb{R}^K_{\geq 0}$ and let $\Omega$ be the full domain of $Z$. Using the same $\psi$ as above, define the *local inconsistency contrast*

$$\Delta_\psi(S) = \psi\big(\mathbf{M}^{(S)}(h)\big) - \psi\big(\mathbf{M}^{(\Omega)}(h)\big),$$

which reports the *level* of inconsistency in $S$ relative to the full range. One can summarize locality by the collection $\{(S, \Delta_\psi(S), \mu(S))\}$, where $\mu(S)$ is the slice prevalence (probability mass or measure). For a continuous profile, one can define a pointwise version $\ell(z) = \psi\big(\mathbf{M}^{(B_\epsilon(z))}(h)\big) - \psi\big(\mathbf{M}^{(\Omega)}(h)\big)$ using a neighbourhood $B_\epsilon(z)$ (sliding window or kernel); "hotspots" satisfy $\ell(z) > 0$.

**Sensitivity** measures how inconsistency changes under plausible perturbations. Let $\Pi$ be a set of pre-registered perturbations to labeling, features, constraints, or sampling. For each $\pi \in \Pi$, recompute $\mathbf{M}^\pi(h)$ and $\psi(\mathbf{M}^\pi(h))$. One can then evaluate:

$$\text{Sens}_\psi = \max_{\pi \in \Pi} \big| \psi\big(\mathbf{M}^\pi(h)\big) - \psi\big(\mathbf{M}(h)\big) \big|, \qquad \text{Sens}_j = \max_{\pi \in \Pi} \big| M_j^\pi - M_j \big|. \tag{7}$$

---

**Inconsistency width and documentation protocol**

An inconsistency report is adequate if it is:

1. pre-registered (choice of $\psi$ and summaries),
2. calibrated on pilot data with uncertainty quantification,
3. reviewed with affected stakeholders,
4. validated for reliability and construct validity,
5. monitored post-deployment for drift with versioned SPHERE cards.

---

## A.2 Calibration protocol (SPHERE-aligned)

We align band calibration and reporting with the five SPHERE dimensions [74]:

- **What (Subject).** Enumerate the system component and design goals per metric (for example effectiveness, safety, fairness).

- **How (Process).** Pre-register the metric set, aggregation rules, and candidate bands; estimate empirical distributions via pilot runs and bootstrap confidence intervals.

- **Who (Handler).** Involve intended users and domain experts to review band proposals and articulate consequences of false positives or false negatives for each metric.

- **When (Elapsed).** Fix bands before model selection, then specify monitoring cadence for short-term and long-term drift.

- **Robustness (Validation).** Record reliability checks (inter-rater agreement or re-test stability) and validity checks (construct or external validity) for each metric and for the profile as a whole.

# B Mitigating trade-offs and Overhead in Pluralistic Optimization

To advocate for the argument that pluralistic evaluation need not entail prohibitive training cost or unavoidable utility loss, here we list a range of possible treatments that *reduce* apparent trade-offs among preserving inconsistent metrics and computational overhead relative to naïve multi-objective training. We enumerate these methods through the processing and training pipelines and summarize their effects on utility, constraint satisfaction, and cost.

## B.1 Pre-processing and Representation: Low-cost Data-side Repairs

Data-side interventions reduce dependence on sensitive attributes before any model is trained. Reweighing schemes adjust example weights to match a fair target distribution, while optimized pre-processing learns mappings that decorrelate features from protected attributes subject to label-preservation constraints [84, 85]. These transforms typically deliver large improvements on independence-style criteria (e.g., demographic parity) with modest changes in utility, and can improve worst-group error under skewed prevalence. Overhead is a one-off closed-form or convex optimization step, after which standard training proceeds unchanged.

## B.2 Pre-training Invariance: Adversarially Fair Representations

Learning representations that are predictive of labels yet invariant to sensitive attributes can decouple fairness from any particular classifier. Adversarially fair representation learning trains an encoder to minimize task loss while fooling a sensitive-attribute discriminator, yielding features that transfer to downstream models with reduced disparity [86]. Empirically, this reduces group gaps with limited utility cost and supports re-use across tasks. The computational overhead is a modest constant factor (an auxiliary head and loss term) relative to baseline training, often offset by avoiding repeated fairness-specific retraining for each downstream model.

## B.3 In-training

When retraining is feasible and principled control of criteria is required, reductions offer a sample-efficient route. Agarwal et al. cast constraints such as equalized odds and demographic parity into convex surrogates optimized by repeated calls to a cost-sensitive classification oracle [15]. This procedure converges in a small number of outer iterations and reliably finds near-Pareto solutions balancing accuracy and constraint satisfaction. Its computational profile resembles a handful of reweighted trainings rather than a full multi-objective run, providing strong trade-offs at controlled cost.

Another approach embeds fairness directly into the loss via differentiable proxies of (in)dependence, such as the covariance between scores and sensitive attributes [16]. Penalizing these correlations during optimization reduces disparities on independence-style metrics while preserving competitive utility. The added cost stems from computing constraint gradients per minibatch, which is typically a small constant-factor overhead without architectural changes. This method is model-agnostic and integrates smoothly with standard optimizers.

## B.4 Post-processing: Constraint Satisfaction Without Retraining

A practical way to satisfy group fairness constraints after model training is to adjust decision thresholds rather than re-fit parameters. Hardt et al. formulate equalized odds/opportunity as a small linear program over group-conditional score distributions, producing group-specific thresholds or randomized decisions that meet error-rate parity [26]. Because the procedure operates on calibrated scores, it frequently attains parity with negligible average-accuracy loss. Computationally, it avoids retraining altogether; the dominant work is score histogramming and solving a low-dimensional LP, which is orders of magnitude cheaper than optimization of model weights.

## B.5 Optimizer-level Stabilization for Multi-objective Runs

Jointly optimizing utility, fairness, and robustness can induce gradient conflict that slows or destabilizes training. Conflict-aware update rules project or aggregate task gradients to reduce destructive

interference and accelerate convergence. PCGrad projects each objective's gradient away from conflicting components, while CAGrad computes a conflict-averse aggregate direction, improving simultaneous satisfaction of objectives without increasing model capacity [57, 58]. These procedures add lightweight per-step computations (projections or small QPs) that are typically offset by fewer epochs needed to reach a feasible multi-metric profile.

**Takeaway.** Across stages of the pipeline, these methods demonstrate that theoretical impossibilities often overstate practical tensions: *approximate* joint satisfaction is routinely achievable with bounded computational cost. A staged workflow—cheap post-/pre-processing, followed by targeted in-training constraints or robustness-oriented training, stabilized with conflict-aware optimization—yields plural metric profiles with minimal utility loss and manageable overhead.

