# OpenReview forum: "Embracing Contradiction: Theoretical Inconsistency Will Not Impede the Road of Building Responsible AI Systems"
_NeurIPS.cc/2025/Position_Paper_Track — NeurIPS 2025 Position Paper Track_

### Official Review · Reviewer_vD9X · 2025-08-08

**Significance:** 4
**Presentation:** 4
**Rating:** 8
**Confidence:** 4

**Summary:**

The machine learning community is in conflict over which metrics to prioritize during training; oftentimes, optimizing along multiple axes requires sacrificing performance along a single axis. The authors argue that this behavior is a feature rather than a bug. The inconsistencies in these objectives is useful for ensuring that machine learning systems can engage with stakeholders with different values and also acts as an implicit regularizer to prevent overfitting to a single metric. The authors combine concepts from diverse perspectives on semantic regularization, Rashomon sets, and block shortcut features. The authors close by introducing future directions.

**Strengths:**

- The authors craft excellent arguments with almost every claim either supported by relevant literature or logical arguments
- The writing is super engaging that would encourage lots of reflection in RAI research
- I really appreciate how the authors engage with multiple RAI communities and illustrate how these diverse viewpoints converge in this framework

**Weaknesses:**

- I would have liked to see more references to the empirical evidence like user studies or retrospective analyses (if they exist) of the downstream effects of pluralist ML systems vs singular objective systems
- I would love more guidance on how to use these objectives in practice: what considerations and trade-offs should a researcher make in deciding what to do?

**Questions:**

- There exists interpretability work clarifying when there may exist an interpretability vs accuracy tradeoff, specifically sourcing from label noise [1]. Do you have a sense of when such a trade-off might exist with plurality/inconsistencies?
- Are there ever any cases where plurality hurts (e.g., low stakes decisions like predicting existence of a cup)? How should a researcher decide what metrics to use and when is too much inconsistency too much?



[1] https://proceedings.neurips.cc/paper_files/paper/2023/file/0a49935d2b3d3342ca08d6db0adcfa34-Paper-Conference.pdf

**Alternative Position:**

Yes, and alternative positions are well-considered and addressed by the argument

**Author Identification:**

No.

**Context:**

4

**Discussion:**

4

**Ethics:**

["NO or VERY MINOR ethics concerns only"]

**Position:**

Yes, the paper argues for or against a position related to machine learning.

**Support:**

4

**Thoroughness:**

4

---

### Official Review · Reviewer_JqYS · 2025-08-10

**Significance:** 3
**Presentation:** 2
**Rating:** 5
**Confidence:** 3

**Summary:**

The paper focuses on the tradeoffs and competitions between various trustworthy/ responsible AI metrics. The authors argue that this competition between different metrics is not a problem to be solved. The authors argue that representing diverse moral values and stakeholder priorities. They formalise two types of inconsistencies: intra-concept and inter-concept, and outline why they are valuable. The authors recommend practice-driven theory for such tradeoff balancing.

**Strengths:**

- The topic is quite important for responsible AI and for different optimization strategies for such metrics.

- Maintaining Pareto front of different metrics is a valid suggestion and a very practical solution when many methodologies for maintaining tradeoffs fail.

- The paper is well structure and intuitive to follow.

**Weaknesses:**

- Would have appreciated some guidance on what acceptable inconsistency thresholds were for some of the tradeoffs, more practically?

- Some methods have been able to mitigate tradeoffs between conflicting metrics. The paper doesn't recognise or discuss them.

- The formulation could be naive since it misses out on some aspects of responsible ai, such as explainability, accountability or adversarial attack, where the user's intent is malicious.

**Questions:**

- What framework do the authors envision can help decide the tolerance band?

- Are there certain levels or types of inconsistency that should always be considered unacceptable? For instance, is a model that is equitable for one group but grossly inequitable for another still within the bounds of "acceptable inconsistency"?

- How would this be applied to different aspects of responsible ai?

**Alternative Position:**

Yes, and alternative positions are well-considered and addressed by the argument

**Author Identification:**

No.

**Context:**

3

**Discussion:**

3

**Ethics:**

["NO or VERY MINOR ethics concerns only"]

**Position:**

Yes, the paper argues for or against a position related to machine learning.

**Support:**

3

**Thoroughness:**

3

---

### Official Review · Reviewer_7JLH · 2025-08-13

**Significance:** 3
**Presentation:** 4
**Rating:** 6
**Confidence:** 3

**Summary:**

This paper argues that theoretical inconsistencies among Responsible AI (RAI) metrics (such as conflicting fairness definitions or trade-offs between accuracy and privacy) should be embraced rather than eliminated. The authors propose that such inconsistencies preserve value pluralism, capture richer conceptual information, and act as implicit regularizers that improve model robustness. They recommend shifting from eliminating inconsistency to defining acceptable inconsistency thresholds, documenting normative assumptions, and designing pluralistic evaluation tools.

**Strengths:**

The paper presents a clear conceptual framework distinguishing between intra-concept and inter-concept inconsistencies. It offers a multi-dimensional argumentation for retaining metric inconsistency. The use of concrete examples from fairness, political neutrality, and accuracy–privacy trade-offs makes the argument accessible and grounded. Recommendations are actionable, covering theoretical development, stakeholder engagement, and transparency documentation.

**Weaknesses:**

The paper’s emphasis on embracing inconsistency could benefit from more detailed risk analysis of when inconsistencies may harm stakeholder trust or decision clarity. While mechanisms for leveraging inconsistency are explained, there is limited discussion on the computational and operational costs of implementing multi-metric optimization. The argument sometimes leans on philosophical justification without equally rigorous empirical evidence across diverse deployment contexts. The proposal for “Metric Provenance Sheets” is promising but lacks concrete implementation guidelines or evaluation results.

**Questions:**

1. One can also say that we should not evaluate the ethics of a machine learning model with metrics at all. What would the authors responed to this?
2. How should practitioners determine the optimal “acceptable inconsistency threshold” for high-stakes applications?
3. What governance structures or oversight bodies could ensure multi-metric evaluations remain transparent and accountable?
4. Are there domains or contexts where metric inconsistency might systematically disadvantage certain stakeholders?
5. How might the proposed pluralistic evaluation approach integrate with existing regulatory frameworks that currently rely on single-metric compliance?

**Alternative Position:**

Yes, and alternative positions are trivial straw-man arguments

**Author Identification:**

No.

**Context:**

2

**Discussion:**

4

**Ethics:**

["NO or VERY MINOR ethics concerns only"]

**Position:**

Yes, the paper argues for or against a position related to machine learning.

**Support:**

3

**Thoroughness:**

3

---

### Note · Authors · 2025-08-25

**1-11 Submit Again:**

Probably yes

**1-1 Submission Process:**

4

**1-4 Interest:**

["Panel discussions with other position paper authors", "Structured debates on controversial topics", "Workshops for developing position papers", "Mentorship programs for early-career researchers"]

**1-5 Thoughtful:**

7

**1-6 Supportive:**

9

**1-7 Technical Aspects Versus Position:**

8

**1-8 Gate Keeping:**

6

**1-9 Camera Ready Changes:**

After the final acceptance, we will expand the camera-ready version in three main directions. First, we will deepen the risk analysis, including computational trade-offs between multi-objective optimization (MOO) and single-objective optimization (SOO), highlighting their respective costs and risks. Second, we will strengthen the paper with empirical evidence to illustrate the practical implications of our framework. Third, we will incorporate Metric Provenance Sheets [arxiv.org/pdf/2504.07971] to improve transparency and traceability of the metrics used. Finally, we will discuss methods for mitigating trade-offs, outlining possible strategies for balancing competing objectives in practice.

**3-1 Review Response1:**

7JLH

**3-2 Reaction To Review1:**

Q1: Metrics are imperfect if treated as truths, but abandoning them removes comparability and accountability. Since decisions always involve quantification, we advocate pluralistic use: evaluate with diverse metrics, apply tolerance bands, document assumptions, and pair numbers with qualitative methods so metrics serve as guardrails, not oracles.

Q2: Tolerance bands must be context-dependent. High-stakes domains like healthcare demand minimal disparities, while entertainment can tolerate more. Thresholds should be calibrated through participatory methods (e.g., workshops, audits), with future work systematizing them and ensuring performance is not unduly degraded.

Q3: Governance should combine independent audit panels, evaluation cards that state each metric’s scope and assumptions, and public dashboards that surface trade-offs. This allows oversight bodies and the public to monitor systems holistically and prevents over-reliance on any single metric.

Q4: Intra-concept inconsistencies rarely disadvantage groups, as optimizing plural metrics better preserves representation than narrowing to a consistent subset. Inter-concept inconsistencies risk sacrificing performance, but Rashomon-set studies show plurality often costs little accuracy. Where utility and cost are low, plurality should be enforced.

**3-3 Review Response2:**

JqYS

**3-4 Reaction To Review2:**

Respond to Weakness 1: We appreciate this point. In Section 4, we identify defining tolerance bands as a key research agenda. We avoided fixed thresholds, since they must vary by domain, stakeholder priorities, and legal constraints: healthcare may allow minimal disparities, while recommendation systems may tolerate more. We propose participatory calibration (e.g., workshops, audits), with future work systematizing guidelines. Thresholds should also be robust, ensuring inconsistency reduction does not significantly degrade overall performance.

Respond to Weakness 2: You are right that methods such as post-processing (Hardt et al.), feature engineering (Wick et al.), and causal-path fairness (Li et al.) can reduce tradeoffs. We noted this in Section 2 but will clarify that these results complement our view: theoretical impossibilities overstate practical tensions, and mitigation methods show approximate consistency is often achievable. Keeping plural but inconsistent metrics addresses “gross inequity”: even if inequities cannot be fully eliminated, optimizing across plural metrics generally yields lower inequity than restricting to a consistent subset.

Respond to Question 3: We focused on fairness, privacy, and related tradeoffs because their “impossibility theorems” illustrate our argument clearly. But we agree Responsible AI also includes explainability, accountability, and robustness, each showing inconsistency. For example, explainability involves accuracy–interpretability tradeoffs and differing definitions of “explanation”; accountability varies across Model Cards, Data Statements, and domains; and robustness efforts often reduce predictive accuracy. These cases reinforce our thesis: pluralistic evaluation, with documentation of assumptions and thresholds, should extend across all Responsible AI domains.

**3-5 Review Response3:**

vD9X

**3-6 Reaction To Review3:**

Q1. Increasing the number of metrics complicates interpretability, as trade-offs become harder to disentangle and metrics are rarely independent. Joint optimization obscures the extent to which each metric drives model behavior, and simply reporting performance on individual metrics offers limited insight. A promising research direction is to develop methods that reveal how different metrics influence optimization dynamics during training, providing both theoretical and practical value for understanding their impact.

Q2. Indeed, enforcing plurality on low-stake decisions seems to be less necessary compares to a single, clear metric, and may resulted in sacrifices in predictive accuracy (i.e: inter-concept trade-offs). Yet several recent works on the Rashomon set (e.g: Bell et al. 2023, Dai et al. 2025) indicate that cases that plurality might not be as pricy as previously imagined: within only a minimal level of accuracy loss, one could still find models that meaningfully preserves plurality through optimization algorithms in log-linear time. For tasks that plurality only brings a minimal cost, plurality should be enforced.

---

### Meta-Review · Area_Chair_h1uY · 2025-09-12

**Rating:** 7
**Confidence:** 4

**Strengths:**

The paper argues that inconsistencies in different metrics we optimize for should be reported and treated as a resource. The arguments are well structured and interesting to read, with connections to established concepts such as Pareto curves and the Rashomon Effect. There are also practical recommendations, including defining tolerance thresholds, documenting normative assumptions, and developing pluralistic dashboard.

**Weaknesses:**

While the value of inconsistency is well argued, the risks surrounding stakeholder trust, confusion in regulatory contexts, or difficulties in operationalizing pluralistic metrics, are not discussed in depth. Among different RAI metrics, fairness and privacy are emphasized, while concepts like explainability and adversarial robustness receive little attention. The authors did agree to address some of the concerns raised by reviewers and improve the paper.

**Questions:**

What is the authors’ position on whether ethics in machine learning can or should be evaluated through metrics, and to what extent such evaluation is possible or meaningful?

How could the proposed evaluation framework be integrated with regulatory systems that currently rely on single-metric compliance?

**Ethics:**

There are no ethical concerns.

**Thoroughness:**

2

---

### Decision · Program_Chairs · 2025-09-26

Accept